# Is Social Media Spreading Misinformation on Exercise and Health in Brazil?

**DOI:** 10.3390/ijerph182211914

**Published:** 2021-11-13

**Authors:** Moacir Marocolo, Anderson Meireles, Hiago Leandro Rodrigues de Souza, Gustavo Ribeiro Mota, Dustin Jay Oranchuk, Rhaí André Arriel, Laura Hora Rios Leite

**Affiliations:** 1Department of Physiology, Institute of Biological Sciences, Federal University of Juiz de Fora, Juiz de Fora 36036-900, Brazil; meireles726@gmail.com (A.M.); hlrsouza@gmail.com (H.L.R.d.S.); rhaiarriel@bol.com.br (R.A.A.); laurahrl@gmail.com (L.H.R.L.); 2Exercise Science, Health and Human Performance Research Group, Department of Sport Sciences, Institute of Health Sciences, Federal University of Triangulo Mineiro, Uberaba 38025-180, Brazil; grmotta@gmail.com; 3Sports Performance Research Institute New Zealand, School of Sport and Recreation, Auckland University of Technology, Auckland 1142, New Zealand; dustinoranchuk@gmail.com

**Keywords:** fitspiration, fitness, Instagram, nutrition, public health

## Abstract

Instagram (IG) reaches millions of people, sharing personal content and all kinds of information, including those related to exercise and health. However, the scientific quality of the posted information is questionable. Thus, this study aimed to analyze whether exercise and health information posted by popular Brazilian IG influencers has technical-scientific accuracy. A personal IG account was created to identify Brazilian IG profiles. The inclusion criteria of the accounts were: (1) having 50% of all the shared posts related to topics about exercise and health, such as nutrition, health and wellness, medicine, or physical fitness; and (2) having over 100,000 followers. Qualitative analysis revealed a low quality percentage (38.79 ± 25.43%) for all analyzed posts. Out of all the posts, only 13 (~2.7%) cited a reference endorsing the information. Moreover, the higher quality-ratio score of the posts was not directly associated with the higher educational qualification of the influencers (r = 0.313; *p* = 0.076). Nevertheless, the number of followers was inversely correlated with the educational qualification of the influencers (r = −0.450; *p* = 0.009), but not with the quality-ratio score of the posts (r = −0.178 *p* = 0.322). We conclude that prominent Brazilian IG influencers disseminate low-quality information about exercise and health, contributing to the wide-spreading of misinformation to millions of followers.

## 1. Introduction

In recent years, obtaining high-quality technical information has expanded from scientific conferences or college classes to web-based resources, making these tools extremely popular [1]. The low cost and potential high reach make web-based channels suitable to obtain prompt information, likely explaining their popularity.

Social media platforms (SMPs), such as Facebook, Twitter, and Instagram (IG), has become part of people’s daily life. Facebook has more than two billion active users, while IG reaches one billion users, being the second most popular among young people. Seventy-one percent of its users are 34 years old or younger, compared to 22% and 40% of Facebook and Twitter users, respectively [2].

IG, a SMP exclusively dedicated to posting and sharing pictures and videos attached to brief texts as captions, allows users to easily create personal profiles to spread their own pictures and private information and interact with other users (i.e., followers) easily. The followers receive real-time notifications from the influencer’s accounts [3].

Initially, the primary purpose of these SMPs was to keep in touch with friends and family [2,4], or to provide access to personal information of celebrities [2]. However, the focus of these SMPs has been evolving to include numerous disclosures, including exercise and health interventions. Such a scenario favored IG because of its emphasis on visually appealing images [5]. In this context, the term “Fitspiration” (merger of the words fitness and inspiration), which refers to images, videos, and texts posted to motivate people to exercise or adopt a healthier lifestyle, has gained popularity [6]. Fitspiration’s primary focus is on propagating information related to the benefits in health and well-being by encouraging healthy eating and exercise practice [7]. Fitspiration is found on an extensive range of sites and SMPs, being most common on IG. It allows the users to “tag” their posts with identified words, enabling easy content searching [7]. A quick search for the hashtags “Fitspiration,” and its shortened form ‘Fitspo’, returns over 90 million posts on IG.

Although Fitspiration and its related terms intend to promote positive social influence regarding exercise and health, several aspects about the technical quality of the information posted by the influencers can be questioned. One important fact is that many shared posts seem to hold simply on an influencer’s mere opinion, experience, or marketing, which is frequently not supported by scientific evidence. Combined with a physique-shaped image, enviable lifestyle publicity, and targeted communication techniques, this set of actions creates a codependent relationship between influencer and follower [7]. In addition, influencers encourage the consumption of supplements or form-fitting branded sportswear as a requirement to achieve the ideal body, conveying the impression that appearance is the key factor to success and happiness [8]. These actions, instead of promoting health, may become detrimental to the follower’s life.

The dissemination of misinformation related to Fitspiration has become widespread in a very fast way [9]. However, there is no standard tool to assess the accuracy and reliability of these disseminated contents. Considering that low-quality information may influence followers to adopt damaging behavior regarding exercise and health [8], and consequently, increase the risk of potential health damage, evaluating the quality of SMP information becomes relevant. Although SMPs have a massive worldwide reach, there is limited evidence about the scientific integrity of the information. Thus, this study aimed to analyze whether exercise and health information posted by main IG Brazilian influencers have technical-scientific accuracy. We hypothesized that the shared information in the posts lacks technical-scientific credibility, contributing to the promotion of misinformation.

## 2. Materials and Methods

### 2.1. Ethics Approval and Consent to Participate

According to international web data regulations and Instagram data policy and after consulting the local ethics committee, this study did not require ethics approval, as only secondary unidentified data from open social media were analyzed. No human studies were carried out, and only publicly accessible sources were used. Personal or person-related content was rendered anonymous so that the identification of the subject was not possible.

### 2.2. Account Selection Criteria

Thirty-three accounts were included in the study. A list with 50 IG accounts, presented on an ordinal scale according to the total number of followers, was provided by a specialized online digital research database service (July 2018). The database provides a list of influencers, ordered by number of followers, and hashtags can then be searched. From this search, 33 accounts with the highest number of followers who also used the relevant hashtags fulfilled the inclusion criteria. Accounts publishing fashion, fitness, and health products for marketing purposes were discarded. Account selection criteria included information published only in Portuguese language and the account´s name or description with at least one of the keywords derived from the topics exercise and health: nutrition, health and wellness, medicine, and/or physical fitness. Only those sharing posts mentioning tips, instructions, programs, protocols analysis, or practical information were selected from this list of accounts. Duplicate ones (with the same administration but with different account names) and accounts with posts related to selling (e.g., clothing, supplement, equipment, or products) were excluded. Considering influencers’ marketing statistics, accounts with less than 100,000 followers are classified as micro-influencers. As we proposed to investigate data from the most influential accounts (regarding follower counts), we chose relevant accounts with 100,000 followers and more [8]. From 33 included accounts, 25 were administrated by males and 8 by females.

A personal IG account was created to access the content of the analyzed posts of each IG account included in the study. Data were collected between August and December of 2018. Based on original information published by each account in their stories, with a mean of 1,114,333 followers, the average reach of the content (during 30 days) was ~133.5 million, and the interactions were ~109.8 million.

### 2.3. Technical Evaluation of Influencers

To verify the digital influencers’ qualification and whether they were qualified or authorized to provide information on exercise and health, the following characteristics were checked: (a) academic or professional qualification: educational and professional degrees were contained on the Google website, and the Lattes platform (a Brazilian database that provides curricular information hosted by the National Council for Scientific and Technological Development and Coordination for the Improvement of Higher Education Personnel). In case such information was not found in either database, the account´s administrator was labeled as “without professional qualification”; (b) divergence between academic/professional qualification and the specific content of the published information (e.g., a nutritionist posting about exercise prescription) was also reported. We observed that all influencers uploaded at least one post outside the knowledge area of their academic/professional qualification from this analysis. The academic and professional degrees were classified as: B.Sc., Bachelor of Science; M.Sc., Master of Science; D.Sc./Ph.D. Doctor of Science/Doctor of Philosophy.

### 2.4. Post’s Inclusion and Classification

Figure 1 shows the process of stratifying the posts. Each IG post was allocated into four knowledge areas based on the exercise and health topics: (1) nutrition, (2) health and wellness, (3) medicine, and (4) physical fitness. Within areas, the posts were then classified according to 3 different purposes: (1) teach (i.e., posts that approached “how to use,” “how to execute/perform,” “how many doses to administrate”), (2) inform (i.e., posts that introduced explanations, descriptions, and/or general information about a topic), and (3) comment (i.e., posts that presented technical opinions about a specific topic or thematic). The different types of posts were also separated according to the publication´s format: (a) picture-image only; (b) video only; (c) text only (slide in the format of the picture containing only written text); (d) picture-text; (e) video-text. The legend of each post was not considered, for the stratification of the different types of posts, but was included in the quality-criteria analysis. Based on the above criteria, information from the 15 most recent posts from each influencer (publication time-window of these posts was about three weeks, as influencers tended to upload one post almost every day) was included in the study´s analysis, totaling 495 posts. Although the collection interval was three weeks, they did not happen simultaneously for all influencers. Thus, the total collection period was 5 months. Posts about digital influencers’ personal lives or those that did not fit the knowledge areas were not included in the analyses. The total number of likes of each post were also registered. On IG, views and likes are both used as tools to measure a post´s popularity moving forward. Likes are counted only when the user taps the post. Views refer to posts’ visualization. However, for the analyses, total likes and not views were considered.

### 2.5. Quality Criteria Assessment

A qualitative analysis (Table 1) of all selected posts and accounts was carried out based on four quality criteria (QC) adapted from the previous work [10]. For the analysis, scores ranging from 0 to 4 were assigned to all analyzed posts and accounts. All analyses were verified by two evaluators separately. Their decision based on the 4 QC had to agree; otherwise, a third evaluation was carried out by another appraiser. The inter-evaluators’ agreement for each QC was calculated using the Cohen Kappa statistic (K) and standard error (SE). The Kappa statistic is based on the following equation: K = (PO − PC)/(1 − PC), where PO is the number of observed agreements and PC is the number of agreements expected by chance. The inter-evaluators’ agreement (K ± SE) was (1.0 ± 0.0), (1.0 ± 0.0), (0.84 ± 0.1), and (0.73 ± 0.1) for QC1, QC2, QC3, and QC4, respectively.

In addition, a quality-ratio score (QRS) was proposed to establish a general framework for all accounts. This score was calculated based on the four QC, considering the 33 accounts and the 15 included posts per account. Each account could total a maximum of 20 points. Specifically, the QC1 represents the academic/professional qualification of the influencers, with scores ranging from 0 (without professional qualification) to 4 (PhD formation). QC2 and QC3 are dependent on each other, where we scored one point for those accounts that presented its source of information, but only if the post’s statements were in agreement with the cited references. If this condition was not true and the statements were made in disagreement with the cited source or if the cited source was completely irrelevant to the mentioned topic, one point was removed from the total. QC4 was scored according to the post’s guidance, requiring that the recommendation must be methodologically reasonable/feasible (15 possible points). A function was constructed to make this logical comparison between all QC as follows Equation (1):QRS = [(QC1 (0 − 4) + (+1; −1; 0) + QC4) × 100]/20(1)
considering: +1 when QC2 = QC3; −1 when QC3 < QC2; 0 when QC2 = 0.

### 2.6. Statistical Analysis

Descriptive statistics were performed using Microsoft Excel (Version 2016, Microsoft, Inc., Redmond, WA, USA). The percentage of all posts was calculated concerning the major knowledge areas. Additionally, the normality of the data about the academic and professional qualification, the quality of the posts, the number of followers, and likes was verified using the Kolmogorov–Smirnov test. Spearman´s bivariate correlation test was performed for the determination of correlations among variables. The above-mentioned correlations were classified according to Hopkins’s criteria (www.sportsci.org Accessed on: 23 March 2021) as follows: <0.1, trivial relationship; 0.1–0.3, weak; 0.3–0.5, moderate; 0.5–0.7, strong; 0.7–0.9, very strong; >0.9, nearly perfect. IBM SPSS statistical software (Version 23, IBM Corp., Armonk, NY, USA) was used to perform data analyses. The level of significance adopted was *p* ≤ 0.05.

## 3. Results

Thirty-three IG accounts were included in the analysis, reaching an average of 30 million followers. Using a post hoc statistic power test (with 33 accounts), a power of 0.97 was reached (Statistical test = Correlation: bivariate normal model; Hypothesis 1 = 0.606; Hypothesis 0 = 0; α = 0.05). Fifteen posts per account were analyzed, totaling 495 posts. The posts presented an overall average of 8430 likes. Table 2 presents the total number of posts in parallel with the likes of all five categories of publication formats.

The stratification of all posts according to the four major knowledge areas is depicted in Figure 2. The majority of posts approached nutrition and physical fitness topics. However, while most of the posts about nutrition were informative, the posts about physical fitness had teaching as the main purpose (Figure 2A). Figure 2B shows that informative posts about nutrition caught greater attention in terms of likes. The number of likes was higher in posts about physical fitness when compared to nutrition.

Table 3 shows the academic/professional qualification of the influencers. Notably, 75.8% of all the account administrators had academic/professional qualifications. The analysis also revealed that all influencers uploaded at least one post outside the knowledge area of their academic/professional qualification (i.e., a nutritionist posting information about exercise prescription, data not shown).

The quality criteria analysis revealed that QC1 reached a median of 1 (with scores ranging from 0 (without professional qualification) to 4 (PhD formation)), demonstrating that most of the account´s administrators were academically/professionally qualified to provide information on the mentioned topic (QC1) (Table 4). Nevertheless, out of the 495 posts selected in the study, only 19% (95 posts) cited a scientific reference (QC2). Out of this total (95 posts), only 13 posts cited a reference that endorsed its information (QC3). In terms of viability (QC4), ~44% of the posts shared information scientifically proven viable to be valuable and practical for the general public. Table 4 also shows that the accounts presented a 38.79 ± 25.43% QRS as calculated by Equation (1).

Table 5 presents correlations between followers, number of likes, academic/professional qualification, and QRS. A positive correlation was found between number of likes and number of followers. The academic/professional qualification of the account´s administrator was inversely correlated with number of followers, but not with the QRS. Moreover, no significant correlation was found between number of followers and QRS.

## 4. Discussion

The main objective of this study was to analyze whether information from prominent Brazilian influencers who disseminate information related to exercise and health fields has technical-scientific support. Our main finding was that most IG influencers provide information lacking scientific sources. From the 495 analyzed posts, only 13 (~2.7%) cited a scientific source that endorsed its information. Moreover, even influencers with an academic/professional degree had quality scores as low as ~39%. Nevertheless, these posts´ misinformation directly reaches a large audience of about 30 million people. Thus, this study shows that leading Brazilian IG influencers related to exercise and health share low-quality information, deficient in scientific support, regardless of academic/professional qualifications.

It is important to highlight that ~75% of the investigated IG influencers hold formal academic/professional backgrounds, supposedly capacitating them to prescribe guidelines and, most importantly, to discern the possible adverse effects of the pervasiveness of the posted information. Misinformation may refer to false and inaccurate information [9]. Interestingly, all account administrators evaluated herein shared at least one piece of information beyond the scope of their academic/professional qualifications. This intrusion is possibly one factor contributing to the low quality of the posts and the low quality score obtained by the accounts.

Regarding the quality of the posted information about exercise and health by prominent Brazilian influencers, data from this study revealed that the minority of them were supported by a scientific reference (~2.7%). Such a finding points to the direction that the account administrator’s opinion, experience, or even personal or business marketing is preferentially shared on their accounts instead of scientifically consistent information. It is believed that science is the best way to develop reliable knowledge and provide scientific explanations [11,12]. Therefore, posts whose information is grounded by scientific sources, such as a published scientific journal article or a book, should be perceived as more credible and trustworthy [13]. However, a critical concern about using scientific references was raised in the present study, as only 13 posts cited scientific references endorsing its information. Moreover, 82 posts referred to a reference that did not corroborate the shared information. In comparison, 400 posts did not cite a peer-reviewed source despite appropriate scientific references existing for 217 of said posts. All these characteristics directly contributed to the low quality score obtained by the accounts, as indicated by the lack of significant correlation between the academic/professional qualification of the account´s administrator and the quality of the posts (r = 0.313; *p* = 0.076). Moreover, the last two aspects above seemingly suggest that either the administrators claim something wholly distorted from the reference (e.g., share exercise information while citing a nutrition resource), or they do not care to share reliable information with their followers at all. The reason why so many of these popular influencers risk their credibility by sharing unscientifically based information is a matter of debate; however, the financial benefit of promoting a fake solution or product should not be discounted.

The events mentioned above ultimately support misinformation dissemination as it reaches a large audience, generating apprehension due to its implications to the public [14]. This is reinforced by the fact that the global impact of SMP became large among the population [2]. Therefore, the information provided by its platforms can directly influence the daily routine of a tremendous number of people [2,4]. By following a particular influencer, users may find themselves copying their habits and adopting their advice on a daily basis [8]. In addition, IG provides a sharing option from the source, using another specific smartphone app. Therefore, the number of people who have access to a single piece of misinformation can increase exponentially, in the case of this study, up to more than 30 million people.

Another troubling finding is the negative correlation observed between the number of followers and the influencer´s qualification (r = −0.450; *p* < 0.01), and the low mean-quality-ratio score of 38.79. This evidence points to the direction that the misinformation divulgated by the influencers about exercise and health may negatively impact the quality of life of many people, leading to detrimental behavior [6]. For instance, unsupervised physical exercise can lead to musculoskeletal injuries [15] and exacerbated Fitspiration promulgation may provoke dietary disturbances (bulimia and vigorexia) and mental disorders (depression and chronic anxiety) [6]. One curiosity is that ~44% of the posts shared information scientifically proven to be viable to execute and valuable for the general public. Still, the influencer was not careful enough to cite a reference to endorse it (only 19% of the post included a reference to endorse it; however only 13 out 95—13.6%—used an adequate reference). These data lead to the belief that users may estimate and give credit to an influencer greatly due to their public image as a celebrity or athletic body shape, for example, instead of a trusted source. IG also allows users to like and comment on others’ posts. At the time of data collection, IG still published the number of views and likes of the posts, which allowed us to make a more consistent descriptive-quantitative analysis. After discussions over the impact of the likes and followers feature on users’ social problems and mental health, IG hid the visualization of theses parameters, even though the usernames of people who have liked the post has remained on display [16]. Herein, the posts presented an average of 8430 likes, considered a key parameter of influence. Regardless of the format, all posts received a considerable number of likes, even those that were predominantly images containing only written text. IG popularity among users is attributed mainly to the emphasis on visual content that has been proven to be more appealing to the audience [5]. This is one important aspect as the attractiveness of SMPs is based on a commercial point of view once most of the posts approach fitness full-body, healthy eating, active exerciser, and sexualization [7]. Particularly when dealing with knowledge areas that print a concept of aesthetics, such types of posts usually call more attention of the public, but take the focus off the primary information and detract from the critical sense about it.

Within the four knowledge areas evaluated in this study, the most usual ones were related to the physical image (i.e., nutrition and physical fitness). More specifically, informative posts about nutrition and teaching posts about physical fitness were the most frequently shared. As indicated by the number of likes, the follower’s interests were mainly focused on comments about nutrition and explanations, descriptions, or general information about physical fitness. These findings show that Fitspiration is a relevant topic of great appreciation.

Although we have provided a descriptive analysis, it was impossible to identify the so-called fake followers (ghost followers purchased from third-party providers). The number of likes and comments can also be manipulated with the help of automated chatbots (i.e., a computer program that can hold a conversation with a person, usually over the internet), further limiting stratification. We agree that the study has clear limitations as it is based on Brazilian influencers, which could limit its audience to the local public. Therefore, despite the fact that the conclusions are regional, the current data could draw attention to the problem elsewhere as social media misinformation may also happen in other countries.

## 5. Conclusions

We conclude that prominent IG Brazilian influencers disseminate low-quality information about exercise and health. The majority of these IG profiles present information lacking scientific support, and 75.8% of all the administrators had academic/professional qualifications. Such a conduct probably contributes to the wide-spreading of misinformation to a large audience (i.e., millions of followers). We emphasize that IG is a social media platform that can be used in a positive but also in a detrimental way. Beyond the mandatory requirements for all influencers, social media platforms could establish explicit rules regulating against misinformation and acknowledge those members who fulfill the anti-fake news regulation.

## Figures and Tables

**Figure 1 ijerph-18-11914-f001:**
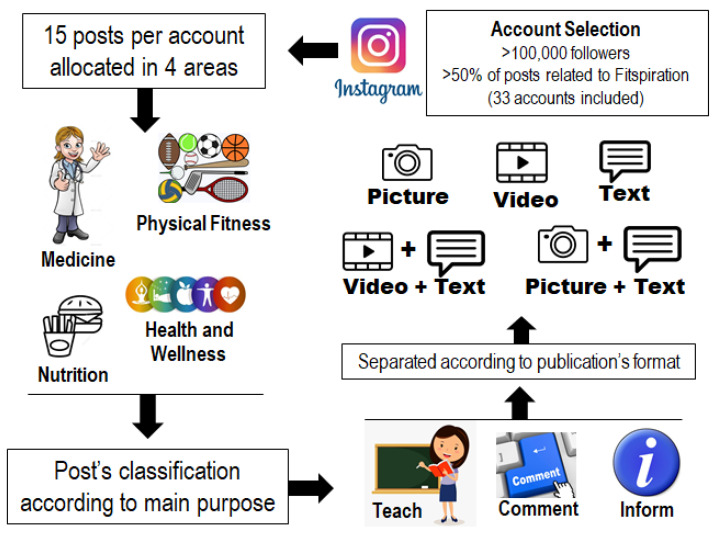
Flow chart of stratification of posts.

**Figure 2 ijerph-18-11914-f002:**
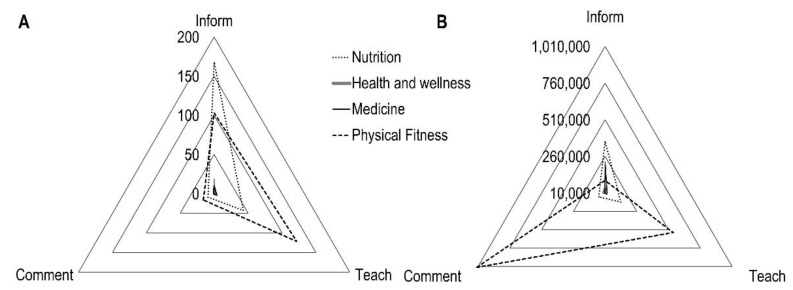
(**A**) Number of posts according to the four major knowledge areas. (**B**) Total number of likes by knowledge area and purpose.

**Table 1 ijerph-18-11914-t001:** Quality criteria assessment performed on the selected accounts and posts.

Quality Criteria	Explanation	Scoring
(1) Is the author academically/professionally qualified to provide information on the mentioned topic?	Account’s administrator must have a compatible academic or professional qualification to be qualified to make recommendations on the mentioned topic (e.g., nutritionists cannot prescribe exercise).	0 points for without professional qualification 1 point for Bachelor of Science 2 points for Specialist 3 points for Master of Science 4 points for Doctor of Science/PhD
(2) Does the author cite any source of information?	Acknowledgment of the source of the information used in the post, such as a scientific study or a book, is preferable for transparency and reliability.	1 point if yes or 0 if not.
(3) Are the post’s statements in agreement with the cited references? ^1^	(1) If a reference is cited, the post must be aligned with its source. Presenting an irrelevant reference should be a problem. (2) If a recommendation is posted, it cannot directly contradict the source (e.g., exercise parameters must be recommended in accordance with the cited reference). (3) If a recommendation is not applicable, the conclusions presented in the post must be based on the cited reference.	1 point only if all three concerns were clearly fulfilled.
(4) Are the post’s guidance supported by any scientific source, even if no reference was cited at all?	The recommendation must be methodologically reasonable/feasible. (1) Even without a scientific source, the recommendation/statement/suggestion cannot be deleterious to health (e.g., advocate the use of steroids). (2) The information needs to be in line with scientifically proven and available evidence on the topic.	1 point only if all two concerns were clearly fulfilled.

^1^ This quality criteria were only pointed if there were any source added for quality criteria 2.

**Table 2 ijerph-18-11914-t002:** Total number of posts and likes according to all five categories of publication’s formats.

	Posts (*n*)	Likes/Post
Picture	196	6392.8
Video	211	12,042.7
Text	2	13,314
Picture/Text	84	4136.1
Video/Text	2	2367.5

**Table 3 ijerph-18-11914-t003:** Academic/professional qualification of IG account´s administrators.

Academic/Professional Qualification	Quantification	B.Sc.	Specialist	M.Sc	D.Sc/Ph.D
Medical doctor	4	-	2	2	-
Nutritionist	10	6	1	1	2
Pharmacist	1	-	-	-	1
Sport Sciences/Kinesiology	9	5	1	1	2
Physiotherapist	1	-	-	-	1
Without professional qualification	8	-	-	-	-

B.Sc., Bachelor of Science; M.Sc., Master of Science; D.Sc./Ph.D., Doctor of Science/Philosophy Doctor.

**Table 4 ijerph-18-11914-t004:** Qualitative analysis of Instagram accounts and selected posts.

QC1 (Account)		QC2 (Post)		QC3 (Post)		QC4 (Post)		Quality-Ratio Score (Account + Post)
Account (*n*)	Score (median)	Posts (*n*)	Score (reached)	Posts (*n*)	Score (reached)	Posts (*n*)	Score (reached)	% (Mean ± SD)
33	1	495	95	495	13	495	217	38.79 ± 25.43

QC, quality criteria. QC1 represents the academic/professional qualification of the influencers, with scores ranging from 0 (without professional qualification) to 4 (PhD formation). QC2 and QC3 represent if the author cites any source of information and if the post’s statements are in agreement with the cited source, respectively. QC4 is related to the post’s guidance, requiring that the recommendation must be methodologically reasonable/feasible. Quality-ratio score was calculated according to Equation (1), where reaching 100% represents maximal scores on all QC.

**Table 5 ijerph-18-11914-t005:** Spearman’s rank correlation among number of followers, number of likes, academic/professional qualification of the influencers, and quality-ratio score.

	Number of Likes	Academic/Professional Qualification	Quality-Ratio Score
Number of followers	0.606 *	−0.450 *	−0.178
Number of likes	-	−0.108	−0.187
Academic/Professional qualification of the influencers	-	-	0.313

* *p* < 0.01.

## Data Availability

The data presented in this study are available on request from the corresponding author.

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
