# Peer review of "Is Social Media Spreading Misinformation on Exercise and Health in Brazil?"

_ijerph, 2021, doi:10.3390/ijerph182211914_

Round 1
Reviewer 1 Report
No major comments to be made the article is well-structured and information has scientific background.
Actually, I do consider that the article can be accepted in the present form as the overall structure and methodology is presented in a clear form.
Some minor modifications could be made in order to improve the overall paper. The comments that I could provide are:
- Study has clear limitations as it is based in Brazil, and it would be difficult to conduct it in other countries. It would be recommend to specify this fact in the title.
- Maybe conclusions could be expanded as no proposals to improve misinformation are made
Author Response
No major comments to be made the article is well-structured and information has scientific background.
Actually, I do consider that the article can be accepted in the present form as the overall structure and methodology is presented in a clear form.
R.: Thank you for the comment.
Some minor modifications could be made in order to improve the overall paper. The comments that I could provide are:
Study has clear limitations as it is based in Brazil, and it would be difficult to conduct it in other countries. It would be recommend to specify this fact in the title.
R.: We have modified the title accordingly. We agree that the study has clear limitations as it is based in Brazil. However, specifying in title the region where the study was conducted could limit its audience to the local public. Therefore, despite the fact that the conclusions are regional, the current data could draw attention to the problem elsewhere since social media misinformation may also happen in other countries.
Maybe conclusions could be expanded as no proposals to improve misinformation are made
R.: Conclusion section was expanded.
Reviewer 2 Report
The topic of the article is very current, it focuses on the negative impact of social networks from an important aspect of human health. The erudition of all seven authors who come from physiological-medical-sports professional workplaces should be appreciated. Thanks to their profession, they are competent in solving research questions about the scientific quality of IG posts regarding healthy nutrition, exercise, etc.
Both the research question and the hypothesis are constructed very well and clearly (l. 74 - 78). The compilation of a research sample also has a high methodological quality. The authors appropriately presented their methodological approach and subsequently applied several quantitative-qualitative statistical methods very precisely. They took into account a number of variables and performed complicated, demanding data collection.
Among the most important and most useful is the QC4 (assessment of possible harmfulness and professional substantiation of posts), where the authors were able to fully apply their erudition and their qualitative subjective approach.
The results are sufficiently interpreted and explained in the discussion and conclusions. The authors also suggest a possible direction for further research (eg the reasons for the willingness of profile administrators to risk their credibility). Such an investigation would require in-depth qualitative methods (eg in-depth interviews, ethnographic participatory observation ...) and would be very interesting.
The text has a high professional level, especially in terms of methodology.However, its reader's friendliness would benefit from giving examples to illustrate the findings.In conclusion, the authors very briefly suggest a few specific situations in which IG can have detrimental effects on people's health behavior.It would be appropriate to give more such specific examples in the text itself (what misinformation appeared in the posts, what they most often concerned, which was contrary to scientific knowledge, which was in line with scientific knowledge ...).
Overall, I rate the text as very high quality and suitable for publication.
Author Response
Overall, I rate the text as very high quality and suitable for publication.
R.: Thank you for the comment.
Reviewer 3 Report
I read with great interest the article titled “Are Social Media Spreading Misinformation on Exercise and Health?” in which the authors concluded that many prominent Brazilian Instagram influencers disseminate low-quality information about exercise and health, contributing to the wide-spreading of misinformation to millions of followers.
Comments and suggestions:
- I would like to suggest reviewing the manuscript for minor grammatical errors and English proofreading.
- Line 56: “A quick search for the hashtags ‘Fitspiration’, and its shortened form ´Fitspo’, returns over 90 million posts on IG.” Please report when was this search carried out.
- Line 63: please define “based-dependency relationship”
- Line 81: Please mention what international regulations were meant in this statement.
- Line 87-88: “was provided by a specialized online digital research database service.” Please mention and cite the database used.
- Line 91: “Accounts publishing fashion, fitness and health products marketing purposes were discarded.” Does that include those who has few sponsored posts, or just those who were dedicated for online marketing?
- Lines 136- 138: “15 posts (collected during approximately three weeks, since influencers tended to upload one post almost every day) from each IG profile were included in the study´s analysis, totalizing 495 posts” Why did you choose the number of 15 posts? Also, it is mentioned above that data collection took place between August and December of 2018. When were those three weeks during which the data collection took place? (The results can be influenced by national events and holidays for example).
- Figure 1: please make sure that none of the used items in the figure is copyrighted and all needed permissions are taken, if applicable.
- Line 155: please mention the full term of the abbreviation SE before mentioning the abbreviation, and present numbers afterwards as (K±SE) instead of K (SE).
- Table 2: I prefer presenting the total numbers of included posts and the number of likes per post. Please remove the middle column (the number of likes (n), as it may be misleading).
- Line 237: please correct the percentage, as 13/495≈6
- Lines 245-247: “supposedly capacitating them to prescribe guide-lines and, most importantly, to discern the possible adverse effects of the pervasiveness of the posted information.” How can they prescribe guidelines when only 13 (~2.6%) cited a scientific source that endorsed its information?
- Lines 267- 270: “Moreover, the last two aspects above seemingly suggest that either the administrators claim something wholly distorted from the reference (e.g., share exercise information while citing a nutrition resource), nor do they care to share reliable information with their followers at all.” Maybe they do care, but they were not trained to use scientific article and evidence-based references to cite for their viewers. We can suggest that having a registered verified page (blue checkmark badge), an influencer must fulfill certain requirements in his or her own field, in addition to mandatory requirements for all influencers.
- Line 293: “Still, the influencer was not careful enough to cite a reference to endorse it.” Please indicate a percentage, as not all of them did not share a valid reference.
- Line 320: please define chat-bots for the readers.
- Please add more limitations for your study. For instance, you did not include whether there was a disclosed conflict of interest by the influencer in your scoring system, did they differentiate between advertisement and content? And so on.
- Line 325: “although their administrators have an academic/professional degree.” Correct it: 75.8% of all the administrators had academic/professional qualifications.
- Please add recommendations and suggestions for Instagram and other social media platform to benefit from your study’s results in improving the quality of their content.
- Please modify the references format according to the journal’s guidelines for authors.
Author Response
I read with great interest the article titled “Are Social Media Spreading Misinformation on Exercise and Health?” in which the authors concluded that many prominent Brazilian Instagram influencers disseminate low-quality information about exercise and health, contributing to the wide-spreading of misinformation to millions of followers.
Comments and suggestions:
- I would like to suggest reviewing the manuscript for minor grammatical errors and English proofreading.
R.: Thank you for the comment. One of the co-authors is native English speaker and revised the manuscript.
- Line 56: “A quick search for the hashtags ‘Fitspiration’, and its shortened form ´Fitspo’, returns over 90 million posts on IG.” Please report when was this search carried out.
R.: The information was included in the manuscript.
- Line 63: please define “based-dependency relationship”
R.: The term refers to relationships where a person is needy, or dependent upon, another person (influencer/social media). To better imply such meaning the term was replaced by codependent in the manuscript.
- Line 81: Please mention what international regulations were meant in this statement.
R.: This section was better described. We have used previous studies for this section (https://doi.org/10.1186/s12889-019-7387-8). Also, information about data policy was found in https://legal.thomsonreuters.com and in the Instagram policy.
- Line 87-88: “was provided by a specialized online digital research database service.” Please mention and cite the database used.
R.: We used the https://www.traackr.com. Since it is a paid service, we chose not to disclose it in the article for the fact of not advertising
- Line 91: “Accounts publishing fashion, fitness and health products marketing purposes were discarded.” Does that include those who has few sponsored posts, or just those who were dedicated for online marketing?
R.: Only accounts that were totally dedicated to online marketing were disregarded. Those who had few sponsored posts were considered, however, these specific sponsored posts were excluded of the analysis.
- Lines 136- 138: “15 posts (collected during approximately three weeks, since influencers tended to upload one post almost every day) from each IG profile were included in the study´s analysis, totalizing 495 posts” Why did you choose the number of 15 posts? Also, it is mentioned above that data collection took place between August and December of 2018. When were those three weeks during which the data collection took place? (The results can be influenced by national events and holidays for example).
R.: We collected data from the 15 most recent posts, where the majority of views were concentrated. Although the collection interval was 3 weeks, they did not happen simultaneously for all influencers. Thus, the total collection period was 5 months. We modified this part of the text for better understanding.
- Figure 1: please make sure that none of the used items in the figure is copyrighted and all needed permissions are taken, if applicable.
R.: No item in the figure needs permission.
- Line 155: please mention the full term of the abbreviation SE before mentioning the abbreviation, and present numbers afterwards as (K±SE) instead of K (SE).
R.: This part was amended (updated lines: 152-156)
- Table 2: I prefer presenting the total numbers of included posts and the number of likes per post. Please remove the middle column (the number of likes (n), as it may be misleading).
R.: The column was removed as requested.
- Line 237: please correct the percentage, as 13/495≈6
R.: The percentage seems to be corrected since 13/495≈2.7.
Lines 245-247: “supposedly capacitating them to prescribe guide-lines and, most importantly, to discern the possible adverse effects of the pervasiveness of the posted information.” How can they prescribe guidelines when only 13 (~2.6%) cited a scientific source that endorsed its information?
R.: Such low percentage of scientific source suggests that influencers simply could not prescribe guidelines on social media without following strict anti-misinformation rules.
- Lines 267- 270: “Moreover, the last two aspects above seemingly suggest that either the administrators claim something wholly distorted from the reference (e.g., share exercise information while citing a nutrition resource), nor do they care to share reliable information with their followers at all.” Maybe they do care, but they were not trained to use scientific article and evidence-based references to cite for their viewers. We can suggest that having a registered verified page (blue checkmark badge), an influencer must fulfill certain requirements in his or her own field, in addition to mandatory requirements for all influencers.
R.: The above-mentioned suggestion was included in the new conclusion
- Line 293: “Still, the influencer was not careful enough to cite a reference to endorse it.” Please indicate a percentage, as not all of them did not share a valid reference.
R.: We agree and this information was included.
- Line 320: please define chat-bots for the readers.
R.: We agree and added this information. Chatbot is a computer program that can hold a conversation with a person, usually over the internet.
- Please add more limitations for your study. For instance, you did not include whether there was a disclosed conflict of interest by the influencer in your scoring system, did they differentiate between advertisement and content? And so on.
R.: Only accounts that were totally dedicated to online marketing were disregarded. Those who had few sponsored posts were considered, however, these specific sponsored posts were excluded of the analysis. We also added more limitations to the study.
- Line 325: “although their administrators have an academic/professional degree.” Correct it: 75.8% of all the administrators had academic/professional qualifications.
R.: The sentence was corrected (updated line: 328).
- Please add recommendations and suggestions for Instagram and other social media platform to benefit from your study’s results in improving the quality of their content.
R.: The conclusion was expanded with a proposal to improve misinformation by social media platforms.
- Please modify the references format according to the journal’s guidelines for authors.
R.: The references have been modified as requested.